# Rethinking Fluid Responsiveness during Septic Shock: Ameliorate Accuracy of Noninvasive Cardiac Output Measurements through Evaluation of Arterial Biomechanical Properties

**DOI:** 10.3390/jpm14010070

**Published:** 2024-01-05

**Authors:** Vasileios Papaioannou, Theodoros Papaioannou

**Affiliations:** 1Intensive Care Unit, Alexandroupolis University Hospital, Democritus University of Thrace, Dragana, 68100 Alexandroupolis, Greece; 2Biomedical Engineering and Cardiovascular Mechanics Unit, 1st Department of Cardiology, Hippokration University Hospital, Medical School of the National and Kapodistrian University of Athens, 11527 Athens, Greece; thepap@med.uoa.gr

**Keywords:** arterial blood pressure, sepsis, fluids, pulse wave velocity, aortic compliance

## Abstract

Beat-to-beat estimates of cardiac output from the direct measure of peripheral arterial blood pressure rely on the assumption that changes in the waveform morphology are related to changes in blood flow and vasomotor tone. However, in septic shock patients, profound changes in vascular tone occur that are not uniform across the entire arterial bed. In such cases, cardiac output estimates might be inaccurate, leading to unreliable evaluation of fluid responsiveness. Pulse wave velocity is the gold-standard method for assessing different arterial biomechanical properties. Such methods might be able to guide, personalize and optimize the management of septic patients.

## 1. Introduction

Severe systemic inflammation caused by infection or trauma may lead to a new continuum of disease: sepsis, septic shock and multiple organ dysfunction syndrome (MODS), which is the leading cause of death in intensive care units (ICUs) [1,2]. Septic shock is an extremely complex condition whose major pathophysiological changes include vasoplegic shock, altered microvascular flow and diffuse endothelial injury [3,4,5]. These changes play a central role in early hemodynamic management of septic patients. Usually, large amounts of fluids and vasopressors are needed in the first hours for hemodynamic optimization. Recent Surviving Sepsis Campaign (SSC) guidelines suggest, based on low-quality evidence, that for patients with sepsis-induced hypoperfusion or septic shock, at least 30 mL/kg of intravenous crystalloid fluid should be given within the first 3 h of resuscitation [6].

However, microvascular leak, profound vasorelaxation and adrenergic hyporesponsiveness may lead to significant tissue edema, acute kidney injury and increased probability of death. For these reasons, the SSC has changed its 2016 statement from a recommendation to a suggestion. Furthermore, an important aspect of fluid administration is to consider withholding or withdrawing resuscitation fluids when they are no longer required. Like for antibiotics, the duration of fluid therapy must be as short as possible.

In this respect, a four-hit model of septic shock was suggested, where four different dynamic phases of fluid therapy were proposed with the acronym ROSE: 1. Resuscitation, 2. Optimization, 3. Stabilization and 4. Evacuation [7]. Each phase requires a different therapeutic attitude in terms of fluid administration. This new paradigm of fluid stewardship aims to answer four basic questions according to the concept of the ‘4 Ds’ of fluid therapy, namely, drug, dosing, duration and de-escalation. The four questions are: when to start fluid administration, when to stop, when to start de-resuscitation and when to stop de-resuscitation. The purpose of such an approach is giving fluids during the first hours of the resuscitation phase and limiting their use during the stabilization phase of septic shock [7].

Based on the SSC 2021 guidelines, a significant tool for more accurate and personalized fluid therapy during sepsis-induced hypoperfusion can be the use of different dynamic instead of static measures of fluid responsiveness, meaning an increase in cardiac output (CO) upon fluid administration, the use of echocardiography, lactate serum levels and capillary refill time as indirect measures of peripheral perfusion [6].

Another reason for limiting large volume administration is that fluids might increase cardiac output but fail to restore low blood pressure (BP) [3,4,5]. In this respect, better understanding of arterial biomechanical properties during dynamic changes in peripheral circulation and vascular tone is of paramount importance for accurate estimation of adequacy of flow and appropriate use of fluids and vasopressors by the carrying physician at the bedside. Thus, in this article we propose a novel research approach for evaluating the effects of compliance of different arterial segments upon CO estimation during septic shock.

## 2. Arterial Biomechanics and Pulse Wave Velocity in Health and Disease

The arterial system has two interrelated hemodynamic functions: 1. arteries behave as pipes to deliver adequate blood flow to the peripheral tissues (a conduit function) and 2. they behave as a ‘windkessel’ (a hydraulic filter and reservoir at the same time) to dampen blood flow and pressure oscillations caused by the intermittent character of left ventricular (LV) ejection and store a percentage of stroke volume during systole, in order to ensure continuous peripheral flow and pressure in a specific pressure range (a dampening function) and during the diastolic phase of the cardiac function [8]. The efficiency of the first property relies on the width of the arterial diameters and the low resistance (R) of large arteries to flow. The efficiency of the second function depends on the elastic properties of the arterial walls and their geometry, including diameter and length. Arterial compliance (C) describes the absolute change in volume (ΔV) due to change in pressure (ΔP) and reflects the ability of arteries to accommodate the volume ejected from the LV instantaneously [8,9].

In clinical practice, the gold-standard method to evaluate C is the measurement of pulse wave velocity (PWV), which is based on the calculation of the pulse transit time (ΔΤ, delay time) of a pressure wave between two arterial sites. The distance between the two sites divided by the pulse transit time determines the wave speed (PWV in m/sec) [8] (Figure 1).

Other indirect methods based on aortic pulse wave analysis have been used to assess wave reflections, which are related to arterial stiffness (1/C), such as the augmentation index (AI) and the reflection time index (RTI) [10]. Briefly, AI reflects the enhancement of aortic systolic pressure due to the return of reflected waves at central aorta, whereas RTI represents the time (Δt) it takes for the pressure waves to travel from the central aorta to the peripheral reflecting sites and return to the aorta within a cardiac cycle [8,10].

Through applanation tonometry, central AI can be computed from peripheral artery pulse waves (i.e., radial and brachial) using validated transfer functions (TFs) (Figure 2).

Since the arterial system is heterogeneous, characterized by a stiffness gradient with progressive PWV increase from the aorta and elastic arteries towards the peripheral muscular conduit arteries, blood pressure waveforms at different sites of measurement reflect regional arterial properties and, eventually, arterial stiffness [8,9,10,11,12,13].

Plenty of evidence now exists indicating that aortic PWV is an independent, strong predictor of cardiovascular risk and mortality in several populations [14]. An increase in arterial stiffness can increase left ventricular load, thus influencing LV function. A reduction in arterial elasticity leads to an increase in pressure wave speed and consequently the arrival of reflected waves during early systole, augmenting peak systolic pressure and pulse pressure. In parallel, coronary perfusion can be compromised due to the lower diastolic pressure.

In addition, the normal stiffness gradient permits the reflected wave to return in diastole due to low aortic PWV. However, with ageing, the aortic stiffness increases to a far greater extent than the peripheral stiffness, limiting and progressively decreasing the partial reflection, something that can increase the transmission of pulsatile energy into the peripheral microcirculation. Such an effect is especially pronounced in the brain and the kidney, two organs highly perfused with low resistance [8]. In this respect, increased aortic PWV has been demonstrated to be an independent predictor of all-cause and cardiovascular death in patients suffering from chronic kidney disease and diabetes, as well as in the general population [8,15].

In ICU patients, only a few studies exist exploring the multi-factorial impact of complex mechanical properties of peripheral and central arteries on patients’ hemodynamics and response to therapy during shock [16,17].

In a recent study including 45 septic patients, arterial stiffness estimated with carotid-to-femoral pulse wave velocity within 24 h of admission to the ICU was higher than in the general population (more than 12 m/sec), while PWV > 24.7 m/sec was associated with shorter survival time [18]. Such findings have been related to functional rather than structural causes, particularly with high levels of vasopressor use.

In another study including 21 Japanese septic patients, the cardio-ankle-vascular index (CAVI), which reflects the stiffness of the whole arterial tree, was measured upon the presentation of patients to the emergency department and one week after sepsis treatment [19]. CAVI was increased after sepsis treatment and was negatively correlated with serum levels of procalcitonin. According to the authors, arterial stiffness is altered depending on sepsis severity, whereas CAVI might transiently decrease at the acute phase of sepsis and rise to original levels after treatment [19]. Such findings have been attributed to sepsis-induced nitric oxide overproduction mainly in muscular arteries, with subsequent progressive reduction in arterial stiffness from aorta to ankle [18,19].

In the next section, we will discuss the limitations of currently used noninvasive methods of cardiac output measurements and their potential impact on clinical decision making regarding fluid and vasopressor use in septic shock patients. Subsequently, we will formulate our hypothesis for a more accurate assessment of hemodynamics based on measurements of different arterial biomechanical properties.

## 3. Limitations of Noninvasive Cardiac Output Measurement during Septic Shock

Beat-to-beat estimates of stroke volume (SV) from the arterial pressure waveform (pulse contour) typically rely on the assumption that changes in the waveform morphology are related to changes in blood flow and central vasomotor tone [9]. According to the simplified Windkessel model of human circulation, aortic pulse pressure (PP) can be estimated as follows [8,20]:PP = (k × SV)/total arterial compliance (1)
where factor k is influenced by body height, heart rate, PWV, arterial compliance and vasomotor tone. In case of stable arterial compliance and PWV, the central arterial PP will vary according to LV output [9,15]. Recently, different commercial devices have evolved aiming at computing CO from the direct measure of peripheral arterial PP via an indwelling arterial catheter. However, such measurements presume a stable PWV and vascular tone [12]. Moreover, fluid-filled catheters, transducer characteristics and frequency response may lead to distortion of the signal where underdamping, overdamping, zeroing and calibration errors can be significant sources of measurement error [20].

Even though vasomotor changes during hemorrhage, hypertension or pharmacologically induced hypotension have been found to occur in a unified way [21], in patients with septic shock, profound changes in vascular tone occur that are not usually uniform across the entire arterial bed, since sepsis may induce pathological vasodilation and decreased adrenergic responsiveness across different segments of the vascular tree [3]. In such cases, the relation between blood pressure and flow in the central arterial compartment could become dissociated in the periphery due to alterations in both central and peripheral arterial biomechanical properties.

In two studies including hemodynamically stable critically ill patients [15] and patients with different causes of shock during volume expansion [16], noninvasively estimated aortic PP through radial tonometry was the same as invasively recorded peripheral PP. However, such findings remain inconclusive regarding hemodynamic monitoring during septic shock, since they are originated from different and heterogeneous study populations.

In an experimental porcine model of acute endotoxic shock, a decrease in aortic and an increase in peripheral arteries’ compliance were found early during fluid resuscitation [22]. Since aortic compliance is pressure-dependent, low BP and subsequent underfilling of the aorta should have reduced arterial stiffness [8]. On the contrary, septic shock induced dissociation in the calculated C from central and peripheral pressure in a way that the central compartment behaved as more stiff (less compliant), whereas the peripheral behaved as if it was more compliant. Such vascular decoupling might significantly decrease the accuracy of different arterial pressure-derived flow devices during profound vasomotor tone changes.

Similar results were found in another recent experimental study, where polymicrobial sepsis induced a severe vascular disarray, decoupling the usual pressure wave propagation from central to peripheral sites [23]. As a result, an inversion of the physiologic pulse pressure amplification took place upon onset of septic shock, with a higher PP in the central aorta than in the peripheral arteries. Another important finding of this study was that such a compromised condition was not resolved by fluid and vasopressor administration.

## 4. Rationale of Our Hypothesis

Different manufacturers have suggested recalibration when clinical conditions rapidly change [9]; however, most of the observed changes in aortic and peripheral compliance occur after the development of significant hypotension and mainly during fluid administration [22]. As a result, it has been proposed that ‘*tracking continuously over time dynamic changes in coupling/decoupling between central and peripheral arterial compartments during significant hemodynamic changes, might be a useful and innovative approach for creating different algorithms in order to match the decoupled state and use it for more accurate estimate of CO, reducing reporting bias*’ [22]. Consequently, someone must know ‘*when the decoupled state occurred, its specific characteristics and when it resolved*’ [22].

Vasoactive drugs might also affect the reliability of peripheral PP to accurately predict changes in central pulse pressure and subsequently CO. Thus, different studies have demonstrated that an incremental infusion of noradrenaline may increase aortic stiffness due to enhanced amplitude and earlier return of wave reflections [24,25]. This is likely to have important clinical implications in septic shock patients. However, no study so far has evaluated the potential effect of vasopressors on arterial stiffness during hemodynamic management of septic shock.

For these reasons, we suggest that central and peripheral arterial biomechanical properties should be evaluated, using noninvasive recordings of arterial pulse waves at different arterial sites, through implementation of gold-standard and validated technology (SphygmoCor CPV System, AtCor Medical Pty Ltd., Melbourne, Australia, http://atcormedical.com/). Furthermore, mapping of elastance at different arterial segments should also be performed. From these measurements, CO can be estimated in septic shock patients, assessing its relationship with values recorded invasively through pulmonary artery (PA) catheters and through peripheral PP recordings.

Another important issue for accurate CO measurement, according to our hypothesis, is its estimation during volume administration.

Reduced accuracy of noninvasive CO estimation has important clinical relevance in decision making regarding fluid administration during septic shock. Fluid responsiveness has been traditionally defined as an increase of more than 10–15% of CO upon administration of 500 mL of saline solution [3]. Fluid and vasopressor administration for achieving a mean arterial pressure of approximately 60–70 mmHg, particularly during the first 3 h of septic shock, remains the cornerstone of hemodynamic management in such patients [4,5,6]. However, many studies have shown that positive fluid balance during the first 2–3 days of hospitalization is related to increased mortality, length of stay in the ICU and cost of treatment [26,27].

As a result, different dynamic indices of volume responsiveness, such as stroke volume variation (SVV) and pulse pressure variation (PPV), have been evaluated and found to predict an increase in CO upon fluid challenge more accurately than static indices, such as central venous pressure or pulmonary arterial occlusion pressure [6,28].

Furthermore, respiratory changes in inferior vena cava (IVC) diameter have been suggested to predict fluid responsiveness in mechanically ventilated patients [29]. In this respect, a distensibility index has been proposed:IVC distensibility index = (IVCmax − IVCmin)/IVCmean (2)

This index evaluates change in IVC diameter with positive pressure. An increase of more than 18% has been found to predict fluid responsiveness with a sensitivity of 90% and a specificity of 100% [29]. However, IVC ultrasound may fail to predict fluid responsiveness in cases of high positive end-expiratory pressure (PEEP), presence of auto PEEP, low tidal volume, abdominal hypertension and right ventricular dysfunction [30].

Moreover, an increase in CO following passive leg raising (PLR) has been shown to be the best predictor of fluid responsiveness [31]. Recently, a combined index named arterial dynamic elastance (E_dyn_) that is the simultaneous ratio of PPV to SVV has been proposed to answer the question: will an increase in flow also increase blood pressure [32,33]? In the case that E_dyn_ is high, arterial pressure will increase if CO increases. On the contrary, if E_dyn_ is low, although CO might increase upon volume administration, arterial pressure will not increase proportionally. As a result, fluid infusion will not be of any benefit, whereas vasopressor administration should be considered to correct hypotension.

In this regard, Guinot and colleagues, in order to predict the decrease in BP in response to norepinephrine dose reduction, studied 35 patients with septic shock [33]. They demonstrated that E_dyn_ was lower in responders (>15% decrease in mean arterial pressure) and that a value less than 0.94 predicted a decrease in BP with a sensitivity of 100% and a specificity of 68%. Nevertheless, the reliability of different dynamic indices and PLR response depends on the robustness with which CO is calculated from peripheral PP recordings, presuming stable arterial resistance. Moreover, because both PPV and SVV are obtained from pulse pressure analysis, a mathematical coupling factor cannot be excluded.

Garcia and colleagues evaluated the prognostic value of E_dyn_ upon arterial pressure response to volume infusion in 53 mechanically ventilated patients after obtaining PPV from an arterial line and SVV from esophageal Doppler imaging [32]. The authors found that a pre-infusion value of E_dyn_ ≥ 0.73 obtained from two independent signals enabled the prediction of arterial pressure response to fluid administration with a sensitivity of 99.9% and a specificity of 91.5%. However, Monnet and colleagues demonstrated that without accurate measurements of aortic diameter, esophageal Doppler monitoring might underestimate fluid responsiveness [34].

In this respect, noninvasive measurement of systolic and diastolic diameters of ascending aorta using echocardiography in the M-mode, at a level of 3–4 cm above the aortic valve from a transthoracic parasternal long-axis view, has been used for the indirect assessment of aortic stiffness [35]. Stefanadis and colleagues found that evaluation of aortic stiffness using different indices derived from such measurements is comparable with invasive methods with a high degree of accuracy [36]. In addition, measurement of changes in aortic circumference is more accurate than measurement of changes in aortic diameter, due to the noncircular shape of the aorta. However, the potential application of such methods in septic patients remains an open question.

Fluid responsiveness should not be confused with fluid dependency and fluid tolerance. Volume expansion is a treatment option in patients exhibiting signs of low peripheral perfusion, such as low CO and BP with increased lactate levels and/or prolonged capillary refill time. In addition, the estimation of the venous side of the circulation can guide fluid therapy using ultrasound markers of venous congestion. In this respect, a new scoring system named VEXUS (venous excess ultrasound score) has been proposed recently, which incorporates hepatic venous and intrarenal venous Doppler and IVC assessment with portal vein Doppler. The presence of a high score has been found to be very specific for the prediction of acute kidney injury following cardiac surgery [37]. In states of right ventricular failure or intravascular fluid overload, the venous compartment becomes congested and can be associated with potential end-organ dysfunction. Nevertheless, its use is operator-dependent and different Doppler signals might be difficult to obtain. In addition, right ventricular dysfunction, stiff lungs, intra-abdominal hypertension or parenchymal renal disease could alter venous waveforms [37].

Recently, a new algorithm named the hypotension prediction index has been developed by industry. It is a machine learning tool that, through evaluation of different arterial waveform features not visible with the naked eye, can predict hypotension, defined as mean arterial pressure (MAP) less than 65 mmHg for at least 1 min, even 15 min before its onset [38]. Although the system was trained and developed in surgical and ICU patients, its external validation was performed only with a surgical patient physiologic data set. Its validity has been tested in different groups of surgical patients with contradictory results [38]. Nevertheless, its potential value in septic patients remains to be evaluated. In any case, the crucial limitation of the algorithm is that it is completely agnostic to the clinical situation and cannot assist the clinician on the required therapy or even if the hypotensive episode can be ignored or requires intervention [39].

In conclusion, and as has been suggested by Magder [40], goal-directed protocols push patients to the plateau of the cardiac function curve, where the normal Frank–Starling mechanism is not operative. On the contrary, a ‘flow-guided responsive’ therapy based on CO as a feedback tool to evaluate response to therapeutic interventions has been suggested to be better than ‘goal-directed’ protocols, since the more reliable CO estimation might improve management of hemodynamics in individual septic shock patients.

Although high CO does not necessarily equate to improved outcomes, we hypothesize that its accurate measurement within the context of complex interrelations between aortic and peripheral compliance may significantly improve the value of different parameters that are currently used at the bedside for optimum fluid and vasopressor therapies in patients with septic shock. In this respect, we suggest that a high accuracy of measurements indicating the similarity of the measured variables to those obtained via the gold-standard method (i.e., a pulmonary artery catheter) is needed when values are used as a ‘trigger’ for either a diagnostic or a therapeutic decision (i.e., a CO increase of more than 12–15% upon fluid challenge as a marker of fluid responsiveness). Thus, the more ill the patient, the greater the need for accuracy, whereas, in less critical states, the trend is the most valuable piece of information and is reflected in the precision of CO device measurements [40].

## 5. Testing Our Hypothesis with Specific Measurements: Mapping of Arterial Stiffness at Different Arterial Segments for More Reliable CO Estimation

Since the major role of volume therapy is to increase cardiac output and subsequently oxygen delivery in septic patients with peripheral hypoperfusion, it is important to know what happened to CO with the therapy. In this respect and in order to override current limitations of noninvasive CO measurement during septic shock, we suggest that an integrated noninvasive monitoring of mechanical properties of systemic arterial circulation in critically ill patients is needed for the assistance of therapeutic interventions.

The specific aims of such an approach would be:1.To describe the trajectories of different arterial biomechanical indices during hemodynamic management of septic shock;2.To evaluate, using statistical models, whether these domains of measurements, in multivariate analyses, predict outcomes of interest (i.e., fluid responsiveness) and therefore identify the incremental predictive value of such ‘physiomarkers’;3.To evaluate, through statistical analyses, how profiles of such indices affect CO measurements.

The ultimate goal would be:1.To guide, personalize and optimize the diagnostic and therapeutic management of ICU patients with septic shock using a wide spectrum of vascular physiomarkers;2.To promptly alert physicians to change their therapeutic strategy based on time variation in these physiomarkers.

The objectives of such studies could be achieved through noninvasive recordings of arterial pulse waves in septic shock patients at different arterial sites (i.e., carotid, radial, brachial or femoral arteries, or at the microcirculation level, such as the finger or toe) using gold-standard and validated technologies (i.e., SphygmoCor, Mobil-O-Graph and Photoplethysmographic-PPGs sensors) [41]. The computation of AI, RTI and PWV will be performed by the software of the aforementioned devices. To detect a potential decoupling between central and peripheral arterial compartments during septic shock, a mapping of arterial stiffness could be performed at different arterial segments by measurement of carotid-to-femoral PWV (aortic stiffness of the elastic type of arteries) and carotid-to-radial and femoral-to-toe PWV (peripheral stiffness of the muscular type of arteries) and during different therapeutic interventions (volume and/or vasopressor administration). The influence of respiration will be minimized, since data will be averaged out over five to ten beats. Finally, estimates of CO values from PP recordings at different arterial segments using existing algorithms [41,42] could be correlated with measurements derived from different commercially available CO measurement devices.

## 6. Conclusions

Pulse wave velocity and aortic waveform analysis are the gold-standard methods for assessing different arterial biomechanical properties. Central and peripheral arterial compliance play a critical role in septic patients, but few studies have already explored their impact on patients’ response to fluid and/or vasopressor therapy, as well as on the accuracy of CO estimation from peripheral blood pressure recordings.

In this respect, different novel indices that have been tested in the literature for accurate estimation of fluid responsiveness, such as dynamic elastance, might underestimate fluid needs, since they fail to consider the role of both aortic and peripheral compliance in noninvasive CO assessment. In addition, potential effects of vasopressor use on arterial waveform characteristics need to be further evaluated through simultaneous measurements of PWV in different arterial segments. In such cases, a potential decoupling between aortic and peripheral compliance could be more precisely evaluated and discriminated from the effects of sepsis on arterial biomechanical properties, allowing a more personalized vasopressor therapy.

Consequently, we suggest that an integrated noninvasive monitoring of mechanical properties of systemic arterial circulation in critically ill, hemodynamically unstable patients should be tested in future studies. Its added value might be to guide, personalize and optimize the diagnostic and therapeutic management of ICU patients, using a wide spectrum of vascular ‘physiomarkers’, and, finally, to promptly alert physicians to change their therapeutic strategy based on time variation in these physiomarkers.

## Figures and Tables

**Figure 1 jpm-14-00070-f001:**
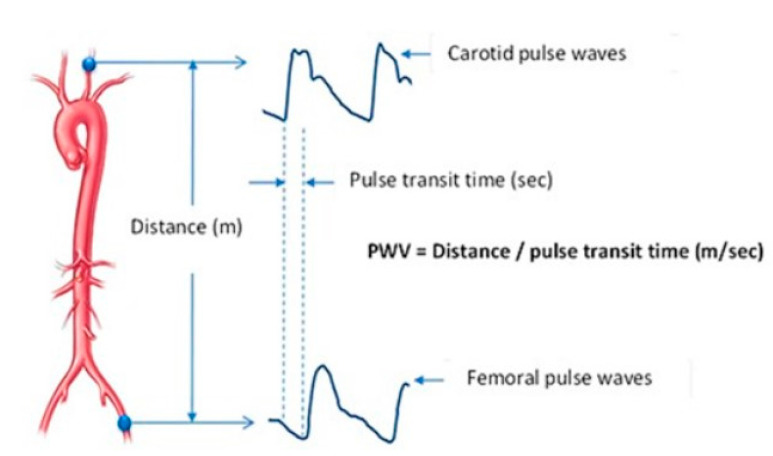
Pulse wave velocity = Length/ΔT. Length = the distance between the two arterial sites (carotid-proximal wave and femoral-distal wave). ΔT = pulse transit time between the two sites.

**Figure 2 jpm-14-00070-f002:**
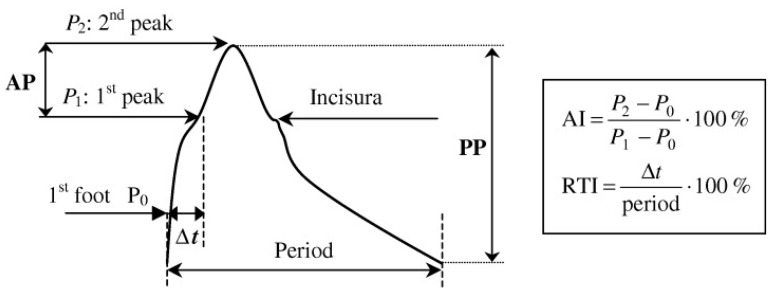
Determination of augmentation index (AI) and reflection time index (RTI) from aortic pressure waves. P_1_: early systolic peak, P_2_: late systolic peak, AP (augmentation pressure): P_2_ − P_1_, Δt: time from onset of pressure wave (P_0_) until return of reflected wave to central aorta (P_1_), PP: pulse pressure (P systolic-P diastolic). Adapted from Ref. [11] with permission from Wiley, 2004.

## Data Availability

Not applicable.

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
