# Peer review of "Rethinking Fluid Responsiveness during Septic Shock: Ameliorate Accuracy of Noninvasive Cardiac Output Measurements through Evaluation of Arterial Biomechanical Properties"

_jpm, 2024, doi:10.3390/jpm14010070_

Round 1

Reviewer 1 Report

Comments and Suggestions for Authors

Papaioannou et al. provided an interesting viewpoint on the accuracy of noninvasive cardiac output measurements for assessing fluid responsiveness in septic shock. The manuscript is well written. However, there are some issues that need to be addressed:

- Line 30-31. Please discuss and cite the recommendations from the 2021 surviving sepsis campaign (doi: 10.1007/s00134-021-06506-y) on the amount of fluids to give in the resuscitation phase.

- Authors should expand the introduction by distinguishing between the resuscitation phase of treating septic shock (where a large amount of fluids is recommended) and the optimization phase, where guidelines recommend to give fluids after careful continuous assessment, thus performed by hemodynamic monitoring, echocardiography, or capillary refill time.

- Line 155. In this context, authors should also cite the role of inferior vena cava collapsibility/distensibility indices in predicting fluid responsiveness with good accuracy, both measured using subcostal or transhepatic echocardiographic approach.

- Does hypotension prediction index have a role in future research on the topic? Please discuss.

Author Response

We thank the reviewer for his/her comments and suggestions

  1. Line 30-31: The recommended suggestions of the reviewer have been added in the revised manuscript. Lines 30-33 & 36-53 of the revised manuscript discuss SSC guidelines on fluid management in different phases of septic shock and proposed methods for fluid responsiveness assessment. Thus, discussion has been expanded according to the reviewer's suggestions
  2. Line 155: In the new section 4, lines 225-233 discuss the issue of IVC along with its limitations for optimum fluid responsiveness assessment.
  3. Hypotension prediction index: Lines 281-291 include discussion and potential limitations of this AI tool for hypotension prediction

Reviewer 2 Report

Comments and Suggestions for Authors

This text discusses the complexities of managing septic shock and multiple organ dysfunction syndrome in intensive care units, emphasizing the importance of understanding arterial biomechanics and the use of pulse wave velocity (PWV) for assessing arterial compliance. The ultimate goal is to guide and optimize the management of ICU patients with septic shock, using various vascular markers, and to enable physicians to adapt their therapeutic strategies based on dynamic changes in arterial properties.

Principal concerns: 

1) Insufficient Emphasis on Practical Implementation: Although the review offers comprehensive theoretical perspectives on the biomechanics of arteries and the treatment of septic shock, the limitations outlined encompass the entire argument put forth in this review. The text concludes with a statement resembling a hypothesis whose genesis is not explicitly explained to the audience. I urge the reviewers to concentrate on this novel aspect of the manuscript and significantly reduce the portion of the limitations that have already been exhaustively described. 

2) It is advisable for authors to refrain from introducing perplexing terminology, such as "flow-directed" strategy. In addition to being incorporated into goal-directed therapy, cardiac output has been deemed ineffective in prior research. It is well understood that in clinical practise, a higher cardiac output does not necessarily equate to improved outcomes. 

3) Pulsatility of blood pressure has multiple physiological implications for organ perfusion (you mention the heart, but other organs may also be present) and is not limited to CO measurement.This particular aspect has not been duly considered. 

4) The review's emphasis on clinical trials and real-world studies that substantiate the theories and methodologies under discussion may be insufficient. Enhancing the reliance on empirical evidence, such as case studies or patient outcomes, would bolster the review's practicality and pertinence in the field of medicine.

Minor points:

Please avoid long paragrpahs . It makes difficult the reading and the message is not clear.

Comments on the Quality of English Language

Good quality of English. 

Author Response

We thank the reviewer for his/her comments and suggestions

  1. We limited significantly the section 3 and divided our hypothesis in two sections, 4 & 5, where we discuss in more detail the rationale of our hypothesis with different examples, as has been proposed by other reviewers, and subsequently, discuss specific aims and methods of applications in section 5. Since the article will be read principally by non ICU physicians, we wanted to present in detail the limitations of currently used methods of CO noninvasive measurement and present our hypothesis as potential alternative, within the context of arterial biomechanics. See lines 59-61, 134-138, 202-211, 298-302. Furthermore, in the conclusions section we added lines 351-359 for a better clarification of our hypothesis in relation with previous discussion.
  2. We definitely agree with the reviewer's comments. However, the term flow-directed therapy originated from Magder (Ref 40) that we cite in the text. Moreover, in lines 292-294, 298-302, 309-313 we discuss the relation between accurate estimation of CO in relation with therapeutic decision making during septic shock.
  3. In lines 108-116 we discuss potential harm of pulsatility upon other organs rather than the heart.
  4.  we have included many new and experimental clinical studies, according to the reviewer's suggestions. See lines 120-133 (section 2), 177-183 (section 3), 242-257 (section 4)
  5. Minor comments: paragraphs have been reduced in size according to the reviewer's suggestions

Reviewer 3 Report

Comments and Suggestions for Authors

Papaioannou and Papaioannou write about a very important topic for patients in septic shock: Fluid responsiveness and cardiac output measurement.

In their viewpoint the authors want to adress the reader's intereset by using a catchy headline "Rethinking fluid responsiveness during septic shock".
Further more by reading "ameliorate accuracy of noninvasive cardiac output measurements" the reader wants to know more about the authors' knowledge since amelioration implies that non invasive CO measurement already works. Well, it does but only by using ultrasound. Unfortunately Ultrasound is not mentioned in the paper.

The section 2. "Arterial biomechanics and pulse wave velocity" is quite long. Of course basic knowledge about this is important but one is waiting for the key questions as mentioned in the headline while reading it.

Even longer is the section 3. "limitations of noninvasive cardiac output measurement during septic shock". Importantly, here the authors describe why non invasive cardiac output measurement cannot work. For readers who are highliy interested in haemodynamics this is something very plausible since deriving an absolute flow (i.e CO) - not a change in flow - from a pressure, even when regarding the physical properties of the vessels, is impossible. As the authors mentioned a change in CO can be estimated by analysing the change of various key points of the pressure wave which represents fluid responsiveness. But this is basic knowledge. 
Further more, when talking about patients in septic shock, these changes become less evident due to the changes in the haemodynamic system as described by the authors.

The key section of the paper 4. "Hypothesis: Mapping of arterial stifness at different arterial segments for more reliable CO estimation" is quite short compared to the others. Also the succus of this section remains vague and does not help to "rethink" fluid responsiveness.

Unfortunately the differentiation between fluid responsiveness and fluid dependency - which is an important issue when evaluating vasopressors vs. volume - is not adressed in this paper. Fluid responsiveness can be misinterpreted misleadingly as indication for volume substitution.

After having read the paper there is no major new topic to rethink fluid responsiveness. I did not get new insights to possibly ameliorate the accuracy of non invasive cardiac output measurements. Anyhow, patients in septic shock need invasive haemodynamic evaluation to guide the appropriate therapy.

The sentence on page 3 (lines 87-89) should be removed since IABP showed no benefit neither short term nor long term in cardiogenic shock patients.

Author Response

We thank the reviewer for his/her comments and suggestions

  1. In section 3 we discuss current limitations of noninvasive CO measurement, particularly in patients suffering from septic shock. Ultrasound examination has been added in lines 225-233 in section 4 and 258-266.
  2. Section 2 headings has been changed (line 62). As the reviewer admitts, in this section we discuss basic concepts of arterial biomechanics and provide many new clinical examples from the ICU, so that the reader, particularly someone who is not an expert in critical care medicine, understands the arterial biomechanical changes that occur in diseases and especially in sepsis (lines 108-133). In lines 134-138 we present the main topic that will be discussed in detail in the subsequent sections.
  3. Section 3 has been reduced significantly. We present current limitations of noninvasive CO measurement along with new examples, derived from the animal literature (line 177-183). The section 4 includes the majority of the discussed literature regarding fluid administation and fluid responsiveness assessment in septic patients, as well as various methodological limitations, thus, formulating the rational of our hypothesis (see lines 202-211, 242-266, 298-302)
  4. Section 4 has been changed to section 5 where we discuss practical implementation of our hypothesis only, along with specific aims. We think that section 4 in the revised manuscript is the most significant one where we highlight the need for accurate estimation of CO and subsequently, fluid responsiveness, in the contect of arterial biomechanical alterations that occur during sepsis. So, we suggest that flow needs to be accurately estimated not for just augmenting its value but for chequing its response to therapy.
  5. Fluid responsivenss vs fluid dependency, as well as fluid tolerance are discussed in lines 267-280 (section 4)
  6. Lines 87-89 of the initial manuscript have been removed. However, Ref 17 in the revised manuscript is still included in the reference list for citing figure 2. 
  7. In general, this is a hypothesis paper and not a review. We try to present another way of thinking about fluid and vasopressos administartion in septic patients and in this respect, we suggest a particular methodology for future research.

Round 2

Reviewer 2 Report

Comments and Suggestions for Authors

I thank the authors for the answers .The made improvements from my point of view are sufficient covering all my concerns.

Reviewer 3 Report

Comments and Suggestions for Authors

In the revised manuscript the authors adressed every point of my review. The manuscript now reads more  clearly and limitations as well as potential benefits of the topic are discussed appropriately.